# Synthesis of Novel Benzo[*b*][1,6]naphthyridine Derivatives and Investigation of Their Potential as Scaffolds of MAO Inhibitors

**DOI:** 10.3390/molecules28041662

**Published:** 2023-02-09

**Authors:** Larisa N. Kulikova, Ghulam Reza Raesi, Daria D. Levickaya, Rosa Purgatorio, Gabriella La Spada, Marco Catto, Cosimo D. Altomare, Leonid G. Voskressensky

**Affiliations:** 1Organic Chemistry Department, Peoples’ Friendship University of Russia (RUDN University), Miklukho-Maklaya St. 6., 117198 Moscow, Russia; 2Department of Pharmacy-Pharmaceutical Sciences, University of Bari Aldo Moro, Via E. Orabona 4, 70125 Bari, Italy

**Keywords:** benzonaphthyridine, activated alkynes, MAO inhibitors, Alzheimer’s disease

## Abstract

In this work, 2-alkyl-10-chloro-1,2,3,4-tetrahydrobenzo[*b*][1,6]naphthyridines were obtained and their reactivity was studied. Novel derivatives of the tricyclic scaffold, including 1-phenylethynyl (**5**), 1-indol-3-yl (**8**), and azocino[4,5-*b*]quinoline (**10**) derivatives, were synthesized and characterized herein for the first time. Among the newly synthesized derivatives, **5c**–**h** proved to be MAO B inhibitors with potency in the low micromolar range. In particular, the 1-(2-(4-fluorophenyl)ethynyl) analog **5g** achieved an IC_50_ of 1.35 μM, a value close to that of the well-known MAO B inhibitor pargyline.

## 1. Introduction

Naphthyridines (pyridopyridines) are nitrogen-containing heterocyclic analogs of naphthalene. Naphthyridines are classified into six types, depending on the location of the nitrogen atoms in the benzene rings [1]. Due to the wide range of biological activities shown by its derivatives, the interest toward the 1,6-naphthyridine nucleus has increased over recent years [2,3,4]. Several functionalized naphthyridines and their benzo/heteroannulated analogs show biological activities, with prospective exploitation as antitumor agents [5,6], mGlu5 receptor antagonists [7], tyrosine kinase SYK inhibitors [8], and antiviral agents [9]. Moreover, benzocondensed naphthyridine is the scaffold of the alkaloid aaptamine contained in the Indonesian sponge *Aaptos suberitoides* [10,11], which is endowed with antibacterial and anticarcinogenic activities [12]. Derivatives of benzo[*b*][1,6]naphthyridine showed antiproliferative (A) and cytotoxic (B) activities against various types of cancer cells [13,14], as well as PDE5 inhibitory activity (C) (Figure 1) [15].

The interest in this bioactive azaheterocyclic moiety prompted us to further explore its reactivity by synthesizing new original derivatives that could possibly be targeted to the treatment of neurological disorders such as Parkinson’s (PD) and Alzheimer’s (AD) diseases [16,17]. Herein, 2-alkyl-10-chloro-1,2,3,4-tetrahydrobenzo[*b*][1,6]naphthyridines (**3**) were prepared and used as starting materials to synthesize novel derivatives, e.g., bearing X-substituted phenylethynyl or indolyl groups at C(1). A 6→8 ring expansion reaction was also applied to compounds **3** to obtain azocino[4,5-*b*]quinoline derivatives.

The biological study combining cheminformatics and biochemical testing focused on AD-related targets. In our ongoing search for biological activities from unconventional chemical scaffolds [18,19], we considered the newly synthesized 1,6-benzonaphthyridine derivatives as worthy of biological investigation. It must be highlighted than piperidine-fused naphthyridine derivatives were previously described by others as dual inhibitors of monoamine oxidase subtypes A and B (MAOs A and B) and of acetyl- and butyrylcholinesterase (AChE, BChE) [20]. This feature is also suggested by the shape similarity of this tricyclic scaffold with that of *β*-carboline, found in the alkaloid harmine and other compounds displaying inhibitory activity toward MAO subtypes [21]. In this study, relying on chemoinformatic predictions, we prioritized the evaluation of inhibitory activity against MAOs A and B.

## 2. Results and Discussion

### 2.1. Chemistry

The synthesis of benzonaphthyridines is mainly based on ring-closing methods, which are used in the synthesis of quinolines. The use of Frindler [22], Pfitzinger [23] and Niementowski [24] reactions, as well as synthesis based on 2-ethynylquinolyl-3-carbaldehydes [25] and aminopyridines [26], makes it possible to obtain derivatives containing functional groups in various positions of the tricyclic system.

Previously, we described conversion of tetrahydrobenzo[*b*][1,6]naphthyridines, which were obtained by the Pfitzinger reaction, to various derivatives under the action of activated alkynes. The structure of the products directly depended on the nature of the substituent in position 10 and Stevens rearrangement products, ylides, 2-vinylquinolines, and benzopyridonaphthyridines can be formed [27,28,29]. However, the mentioned works were mainly focused on substances with acceptor substituents at position 10 because of their easy synthesis by the Pfitzinger reaction. Thus, it was of interest to perform experiments using substances with electron-donating substituents such as chlorine. N-methyl- and benzyl-10-chloro-1,2,3,4-tetrahydrobenzo[*b*][1,6]naphthyridines **3a**–**d** were synthesized by the Niementowski reaction based on condensation of anthranilic acids **1a**–**c** with the appropriate piperidones **2a**,**b** when heated in a phosphorus oxychloride atmosphere. After alkaline treatment of the reaction mass, compounds **3a**–**d** were obtained in the form of yellow crystals with yields of 68–92% (Figure 1).

The study of the reactivity of 10-chloro-tetrahydrobenzo[*b*][1,6]naphthyridines showed that 2-methyl-10-chloro-1,2,3,4-tetrahydrobenzo[*b*][1,6]naphthyridine **3c** was completely inactive in reactions with activated alkynes. Extended boiling and microwave activation in various solvents did not lead to the formation of products, whereas only the initial compound was released back from the reaction mass. However, the presence of an acceptor substituent at C-8 in compound **3d** made it possible to obtain product **4c**. The reaction of 2-benzyl-10-chloro-1,2,3,4-tetrahydrobenzo[*b*][1,6]naphthyridines **3a**,**b** with methyl propiolate proceeded at room temperature in methanol under acid catalysis conditions. The N-vinyl derivatives **4a**,**b** were obtained as a result of debenzylation (Figure 2). We previously described such transformations for N-benzyl-chromenopyridines [30].

We supposed that the functionalization of this system by introducing indole or phenylethynyl fragments onto the tetrahydropyridine ring would significantly expand the potential of tetrahydrobenzo[*b*][1,6]naphthyridines as biologically active compounds and reveal new ways of further achieving chemical modifications. The introduction of a substituent to the nearest position of the nitrogen atom in the tetrahydropyridine fragment was performed by imine salt formation. Such reactions are well described for tetrahydroisoquinolines [31,32,33,34], but they never have been used in case of tetrahydrobenzonaphthyridines. 1-Phenylethynylated benzonaphthyridines **5a**–**h** were obtained as result of the cross-combination of compounds **3** with phenylacetylene in the presence of CuI and diisopropyldiazodicarboxylate (DIAD) (Figure 3 and Table 1).

The nucleophilic addition of benzonaphthyridine tertiary nitrogen to DIAD led to the formation of a zwitterion, which then turned into an iminium salt. The target products **5** were obtained after further alkynylation of the salt with copper acetylenide. The isolation of the reaction products was hampered by the presence of substituted hydrazine in the reaction mixture. This compound was obtained from DIAD and crystalized simultaneously with the target compounds, so it was necessary to use column chromatography.

The structure of compound **5a** was determined by single crystal X-ray analysis (CCDC 2224256, Figure 2).

The phenylethynyl derivatives of benzo[*b*][1,6]naphthyridines **5** turned out to be much more reactive towards activated alkynes. The reaction with methyl propiolate in trifluoroethanol and acetonitrile took place at room temperature after 10 min with the formation of complex separable mixtures. After selecting the reaction conditions, we obtained satisfactory results using subzero temperatures and isopropanol as a solvent. As a result of the interaction of compounds **5d**,**g** with methyl propiolate under these conditions, two products were formed: the Stevens rearrangement products **6a**,**b** and 2-vinylquinoline **7**. The reaction of compounds **5g** and **5d** also yielded quinoline **7**. Under the same conditions, the interaction of compounds **5c**,**d**,**f**,**g** with acetylacetylene led to the formation of products **6c**–**f**, the only products with good yields (Figure 4 and Table 2).

Similarly, the Stevens rearrangement occurred in the case of 1-phenylethynyl-substituted-*β*-carbolines reacted with activated alkynes [35].

The formation of products **6** and **7** started with the Michael addition of nitrogen of the tetrahydropyridine fragment to the activated alkyne leading to the formation of the zwitterion **A**. Then, either a Stevens rearrangement (route **a**) took place with the formation of compound **6**, or further attack on the triple bond of the phenylethynyl fragment yielded adduct **C** (route **b**) and then proton migration and Hoffmann elimination completed this cascade of transformations to give minor product **7** (Figure 5).

Besides phenylethynylation, we introduced an indole fragment at position C1 of benzonaphthyridines **3**. At the first stage, iminium salts were obtained by interaction of benzonaphthyridines **3** with DIAD, and these salts reacted with substituted indoles at the second stage. The reaction was carried out in absolute THF and benzonaphthyridines **8** were isolated by column chromatography. The isolation of the products was hampered by the presence of substituted hydrazine as in the case of phenethynylation. Thus, only compound **8a** was isolated in its individual form, whereas compounds **8b-d** were isolated in mixture with hydrazine **9** (Figure 6).

The interaction of 1-indol-3-yl derivatives of benzo[*b*][1,6]naphthyridines **8** with acetylacetylene in isopropanol led to the expansion of the tetrahydropyridine fragment, with the formation of azocine **10** (Figure 7).

Here, we describe the first example of the formation of tetrahydroazocino[4,5-*b*]quinolines; however, such transformations have been studied for other heterocyclic systems annulated with the tetrahydropyridine fragment and the mechanism of azocine fragment formation has been presented [36,37,38,39,40,41].

### 2.2. Evaluation of Monoamine Oxidase (MAO) Inhibition

Taking advantage of PLATO, a free online platform for structure-based target prediction [42] that relies on a multi-fingerprint similarity search algorithm [43,44], we submitted derivatives **3a**–**d**, **5a**–**h**, **8a**–**b**, and **10a**–**c** for bioactivity prediction. Interestingly, MAOs were found among the targets predicted with higher probability along with binding affinities for dopamine and opioid receptor subtypes only for the N(2)-methyl analogues **5c**–**h** bearing phenylethynyl groups at C1. In contrast, N(2)-benzyl analogs **5a**–**b** and compounds **3**, **8**, and **10** were unpredicted as MAO ligands.

Compounds **5c**–**h** were then tested on human (h) recombinant MAO A and B using previously reported assays [16,17]. The MAO-B-selective inhibitor, pargyline, was used as the positive control. Each compound was first tested at a concentration of 10 μM and then lower scalar concentrations were tested when >60% inhibition was achieved at 10 μM. The IC_50_ values were calculated from the best-fitting inhibition–concentration curves. The MAO A and B inhibition data are summarized in Table 3.

All of the tested compounds showed a certain selectivity toward MAO B, with most of them achieving IC_50_ values in the low micromolar range. The 4-F derivative **5g** showed a noteworthy IC_50_ (1.35 μM), a value that is even lower than that of the reference pargyline. The absence of chemical groups able to create covalent bonds, such as the propargylamine fragment in pargyline, suggested a tight, yet reversible, interaction at the binding site of the enzyme for **5g**. Compounds **5c**–**h** were also assayed as inhibitors of human cholinesterases [45], but they proved to be inactive as AChE inhibitors and scarcely active toward BChE at 10 μM. Meanwhile, the 4-Cl derivative **5h** displayed less than 44% antiaggregating activity against amyloidogenic Aβ(1-40) peptide at 100 μM.

## 3. Materials and Methods

### 3.1. Chemistry

Materials and general procedures. All reagents and solvents were purchased from Merck (Darmstadt, Germany), J.T. Baker (Phillipsburg, NJ, USA), or Sigma-Aldrich Chemical Co. (St. Louis, MO, USA) and, unless specified, used without further purification. The melting points (m.p.) of all of the compounds were determined using a SMELTING POINT 10 apparatus in open capillaries (Bibby Sterilin Ltd., Stone, UK). IR spectra were recorded using an Infralum FT-801 FTIR spectrometer (ISP SB RAS, Novosibirsk, Russia). The samples were analyzed as KBr disk solids and the more important frequencies are shown in cm^−1^. ^1^H and ^13^C NMR spectra were recorded in chloroform-d_3_ (CDCl_3_) or dimethylsulfoxide-d_6_ (DMSO-d_6_) solutions at 25 °C with a 600-MHz NMR spectrometer (JEOL Ltd., Tokyo, Japan). Peak positions were given in parts per million (ppm), referenced to the appropriate solvent residual peak, and signal multiplicities were collected by: s (singlet), d (doublet), dd (doublet of doublets), ddd (doublet of doublet of doublets), t (triplet), q (quartet), br.s (broad singlet), and m (multiplet). MALDI mass spectra were recorded using a Bruker autoflex speed instrument operating in positive-ion reflectron mode (Bremen, Germany). The data of compound **5a** were collected at room temperature using an STOE diffractometer Pilatus100K detector, focusing on mirror collimation Cu Kα (1.54086 Å) radiation, in rotation method mode. STOE X-AREA software was used for cell refinement and data reduction. Data collection and image processing were performed with X-Area 1.67 (STOE & Cie GmbH, Darmstadt, Germany, 2013). Intensity data were scaled with LANA (part of X-Area) in order to minimize the differences in intensities of symmetry-equivalent reflections (multiscan method).

#### 3.1.1. Synthesis of 2-Alkyl-10-chloro-1,2,3,4-tetrahydrobenzo[*b*][1,6]naphthyridines **3a**–**d**

Phosphorus chloride was added dropwise in a volume of 10 mL to anthranilic acids **1a**–**c** (0.0146 mol), which was then added in 1 equivalent excess of (0.0146 mol) 1-alkylpiperidine-4-one **2a**,**b**. Next, the reaction was stirred for 4 h at 100 °C and controlled by TLC in an ethyl acetate—hexane (1:1) system on Silufol plates. After cooling, the resulting solution was neutralized with dilute NaOH solution to pH = 9–10, and the product was extracted with CH_2_Cl_2_. The organic phase was dried over anhydrous sodium sulfate and concentrated on a rotary evaporator. The product was obtained by crystallization from diethyl ether.

**2-benzyl-10-chloro-1,2,3,4-tetrahydrobenzo[*b*][1,6]naphthyridine (3a).** Light-yellow crystals, yield 92%, m.p.= 120–121 °C. ^1^H NMR (600 MHz, CDCl_3_) δ (ppm): 8.16 (1H, dd, *J* = 8.3, 1.4 Hz), 8.00 (1H, dd, *J* = 8.5, 1.1 Hz), 7.69 (1H, ddd, *J* = 8.3, 6.8, 1.4 Hz), 7.55 (1H, ddd, *J* = 8.2, 6.9, 1.2 Hz), 7.42 (2H, d, *J* = 7.0 Hz), 7.37 (2H, t, *J* = 7.6 Hz), 7.31 (1H, t, *J* = 7.3 Hz), 3.93 (2H, s), 3.82 (2H, s), 3.24 (2H, t, *J* = 5.9 Hz), 2.90 (2H, t, *J* = 6.0 Hz). ^13^C NMR (150 MHz, CDCl_3_), δ (ppm): 157.4, 147.5, 139.9, 138.1, 129.9, 129.4(2C), 129.1, 128.8(2C), 127.7, 127.1, 127.0, 125.4, 123.9, 62.8, 54.5, 49.9, 34.0. HRMS (MALDI+) *m*/*z* calcd for C_19_H_17_ClN_2_ in form of [M + H]^+^ ion 309.1159, found: 309.1176.

**2-benzyl-8-bromo-10-chloro-1,2,3,4-tetrahydrobenzo[*b*][1,6]naphthyridine (3b).** Brown crystals, yield 86%. m.p. = 138–139 °C. ^1^H NMR (600 MHz, CDCl_3_) δ (ppm): 8.32 (1H, d, *J* = 2.3 Hz), 7.85 (1H, d, *J* = 8.8 Hz), 7.75 (1H, dd, *J* = 8.9, 2.2 Hz), 7.41 (2H, d, *J* = 7.5 Hz), 7.37 (2H, t, *J* = 7.4 Hz), 7.32 (1H, d, *J* = 7.1 Hz), 3.92 (2H, s), 3.82 (2H, s), 3.21 (2H, t, *J* = 6.0 Hz), 2.90 (2H, t, *J* = 5.9 Hz). ^13^C NMR (150 MHz, CDCl_3_) δ (ppm): 157.8, 145.8, 138.5, 133.3, 130.6(2C), 129.2(2C), 128.6(2C), 127.6, 126.4, 126.1(2C), 121.0, 62.6, 54.3, 49.6, 33.8. HRMS (MALDI+) *m*/*z* calcd for C_19_H_16_BrClN_2_ in form of [M + H]^+^ ion 387.0264, found: 387.0280. The observed characterization data (^1^H) were consistent with those previously reported in the literature [46].

**2-methyl-10-chloro-1,2,3,4-tetrahydrobenzo[*b*][1,6]naphthyridine (3c).** Light-yellow crystals, yield 68%, m.p. = 94–95 °C. ^1^H NMR (600 MHz, CDCl_3_) δ(ppm): 8.18 (1H, dd, *J* = 8.4, 0.9 Hz), 8.00 (1H, dd, *J* = 8.4, 0.6 Hz), 7.70 (1H, ddd, *J* = 8.4, 6.9, 1.4 Hz), 7.56 (1H, ddd, *J* = 8.2, 6.8, 1.2 Hz), 3.84 (2H, s), 3.28 (2H, t, *J* = 6.0 Hz), 2.87 (2H, t, *J* = 6.0 Hz), 2.58 (3H, s). ^13^C NMR (150 MHz, CDCl_3_) δ (ppm): 156.6, 147.2, 139.5, 129.8, 128.9, 126.8, 126.6, 125.1, 123.7, 56.0, 52.5, 46.1, 33.8. HRMS (MALDI+) *m*/*z* calcd for C_13_H_13_ClN_2_ in form of [M + H]^+^ ion 233.0846, found: 233,0831.

**2-methyl-8-nitro-10-chloro-1,2,3,4-tetrahydrobenzo[*b*][1,6]naphthyridine (3d).** Yellow crystals, yield 75%, m.p. = 172–173 °C. ^1^H NMR (600 MHz, CDCl_3_) δ (ppm): 9.10 (1H, d, *J* = 2.5 Hz), 8.43 (1H, dd, *J* = 9.2, 2.5 Hz), 8.09 (1H, d, *J* = 9.2 Hz), 3.88 (2H, s), 3.31 (2H, t, *J* = 5.9 Hz), 2.92 (2H, t, *J* = 6.0 Hz), 2.61 (3H, s). ^13^C NMR (150 MHz, CDCl_3_) δ (ppm): 161.2, 149.4, 146.0, 141.3, 131.1, 128.9, 124.6, 123.5, 121.3, 56.0, 52.2, 46.1, 34.2. HRMS (MALDI+) *m*/*z* calcd for C_13_H_12_ClN_3_O_2_ in form of [M + H]^+^ ion 278.0696, found: 278.0704.

#### 3.1.2. Synthesis of 2-Vinyl-10-chloro-1,2,3,4-tetrahydrobenzo[*b*][1,6]naphthyridines **4a,b** and 1-Vinyl-10-chloro-1,2,3,4-tetrahydrobenzo[*b*][1,6]naphthyridines **4c**

To a solution of 0.2 g of benzonaphthyridines **3a**,**b** in 5 mL of methanol with 0.5 mL of formic acid was added a 1.2 equivalent of activated alkyne. The reaction was kept at room temperature for 10 days. Compounds **4a**,**b** spontaneously fell out of the reaction mass in the form of crystals and were released by filtration.

To a solution of 0.2 g of benzonaphthyridines **3d** in 5 mL of trifluoroethanol with 0.5 mL was added a 1.2 equivalent of activated alkyne. The reaction was kept at room temperature for 15 days. The product was obtained by crystallization from diethyl ether.

**Methyl (2E)-3-(10-chloro-3,4-dihydrobenzo[*b*][1,6]naphthyridin-2(1H)-yl)prop-2-enoate (4a)**. White crystals, yield 59%. m.p. = 171–172 °C. ^1^H NMR (600 MHz, CDCl_3_) δ (ppm): 8.21 (1H, dd, *J* = 8.4, 1.4 Hz), 8.02 (1H, d, *J* = 8.5 Hz), 7.76 (1H, ddd, *J* = 8.3, 6.8, 1.4 Hz), 7.63 (1H, d, *J* = 6.4 Hz), 7.65–7.61 (2H, m), 4.89 (1H, d, *J* = 13.2 Hz), 4.57 (2H, s), 3.71 (3H, s), 3.70–3.69 (2H, m), 3.27 (2H, t, *J* = 6.0 Hz). ^13^C NMR (150 MHz, CDCl_3_) δ (ppm): 169.8, 155.5, 151.4, 147.4, 130.5, 129.1, 129.1, 127.6, 125.3, 123.8, 86.5, 50.9. HRMS (MALDI+) *m*/*z* calcd for C_16_H_15_ClN_2_O_2_ in form of [M + H]^+^ ion 303.0900, found: 303.0911.

**Methyl (2E)-3-(8-bromo-10-chloro-3,4-dihydrobenzo[*b*][1,6]naphthyridin-2(1H)-yl)prop-2-enoate (4b).** Yellow crystals, yield 67%. m.p. =194–195 °C. ^1^H NMR (600 MHz, CDCl_3_) δ (ppm): 8.36 (1H, d, *J* = 2.1 Hz), 7.88 (1H, d, *J* = 9.0 Hz), 7.81 (1H, dd, *J* = 9.0, 2.1 Hz), 7.62 (1H, d, *J* = 13.1 Hz), 4.90 (1H, d, *J* = 13.1 Hz), 4.56 (2H, s), 3.71 (3H, s), 3.69 (2H, m), 3.26 (2H, t, *J* = 6.0 Hz). ^13^C NMR (150 MHz, CDCl_3_) δ (ppm): 169.8, 156.1, 151.3, 146.0, 134.1, 130.9, 126.4, 126.1, 121.8, 86.8, 50.9. HRMS (MALDI+) *m*/*z* calcd for C_16_H_14_BrClN_2_O_2_ in form of [M + H]^+^ ion 381.0005, found: 381.0014.

**(3E)-4-(10-chloro-2-methyl-8-nitro-1,2,3,4-tetrahydrobenzo[*b*][1,6]naphthyridin-1-yl)but-3-en-2-one (4c).** White crystals, yield 62%. ^1^H NMR (600 MHz, CDCl_3_) δ (ppm): 9.13 (1H, d, *J* = 2.2 Hz), 8.49 (1H, dd, *J* = 9.2, 2.5 Hz), 8.13 (1H, d, *J* = 9.2 Hz), 6.82 (1H, dd, *J* = 16.0, 7.0 Hz), 6.04 (1H, d, *J* = 15.9 Hz), 4.86 (1H, d, *J* = 6.9 Hz), 3.41–3.34 (1H, m), 3.27–3.22 (1H, m), 3.16 (1H, dd, *J* = 4.8, 2.4 Hz), 3.03–2.99 (1H, m), 2.60 (3H, s), 2.26 (3H, s). ^13^C NMR (150 MHz, CDCl_3_) δ (ppm): 197.8, 161.2, 149.5, 146.0, 143.0, 140.9, 134.6, 131.0, 128.8, 124.7, 123.9, 121.5, 62.0, 45.6, 42.4, 31.4, 27.7. HRMS (MALDI+) *m*/*z* calcd for C_17_H_16_ClN_3_O_3_ in form of [M + H]^+^ ion 346.0958, found: 346.0981.

#### 3.1.3. Synthesis of 2-Alkyl-10-chloro-1-phenylethynyl-1,2,3,4-tetrahydrobenzo[*b*][1,6]naphthyridines **5a**–**h**

A solution of **3a**,**c** (0.5 g) in 10 mL of THF was cooled to 0 °C, then a 1.5 equivalent excess of DIAD (diisopropylazodicarboxylate) was added. The mixture was stirred at room temperature for 1 h. After cooling it again to 0 °C, a 3 equivalent excess of the appropriate phenylacetylene and CuI catalyst were added. The reaction was stirred at room temperature and controlled by TLC in an ethyl acetate-hexane (1:5) system on Silufol plates. The product was separated by column chromatography.

***2*-benzyl-10-chloro-1-phenylethynyl-1,2,3,4-tetrahydrobenzo[*b*][1,6]naphthridine (5a).** Colorless crystals, yield 24%. m.p. = 139–140 °C. ^1^H NMR (600 MHz, CDCl_3_) δ (ppm): 8.22 (1H, d, *J* = 8.3 Hz), 8.01 (1H, d, *J* = 8.3 Hz), 7.72 (1H, t, *J* = 7.6 Hz), 7.57 (1H, t, *J* = 7.7 Hz), 7.50–7.43 (4H, m), 7.38 (2H, t, *J* = 7.5 Hz), 7.34–7.28 (4H, m), 5.31 (1H, s), 4.09 (1H, d, *J* = 13.0 Hz), 3.91 (1H, d, *J* = 13.1 Hz), 3.35–3.41 (1H, m), 3.31 (1H, td, *J* = 11.5, 3.6 Hz), 3.18 (1H, dd, *J* = 16.6, 3.6 Hz), 3.08 (1H, dd, *J* = 11.6, 6.7 Hz). ^13^C NMR (150 MHz, CDCl_3_) δ (ppm): 156.7, 147.6, 140.6, 138.0, 131.9(2C), 130.2, 129.2(2C), 128.8, 128.6(2C), 128.4(2C), 128.3, 127.6, 126.9, 125.3, 124.2(2C), 122.8, 87.5, 84.3, 59.4, 52.8, 44.8, 33.4. IR spectrum (KBr), υ/cm^−1^: 2223.1 (-C≡C-). HRMS (MALDI+) *m*/*z* calcd for C_27_H_21_ClN_2_ in form of [M + H]^+^ ion 409.1471, found: 409.1483.

***2*-benzyl-10-chloro-1-[(3-methoxyphenyl)ethynyl]-1,2,3,4-tetrahydrobenzo[*b*][1,6]naphthyridine (5b).** Oil, yield 35%. ^1^H NMR (600 MHz, CDCl_3_) δ (ppm): 8.21 (1H, d, *J* = 8.4 Hz), 8.01 (1H, d, *J* = 8.5 Hz), 7.71 (1H, t, *J* = 7.7 Hz), 7.57 (1H, t, *J* = 7.8 Hz), 7.47 (2H, d, *J* = 7.4 Hz), 7.37 (2H, t, *J* = 7.5 Hz), 7.31 (1H, t, *J* = 7.3 Hz), 7.20 (1H, *t*, J = 8.0 Hz), 7.05 (1H, d, *J* = 7.6 Hz), 6.96 (1H, s), 6.86 (1H, dd, *J* = 8.2, 2.6 Hz), 5.30 (1H, s), 4.08 (1H, d, *J* = 13.1 Hz), 3.90 (1H, d, *J* = 13.1 Hz), 3.78 (3H, s), 3.41–3.34(1H, m), 3.30 (1H, td, *J* = 11.6, 3.6 Hz), 3.18 (1H, dd, *J* = 16.4, 3.1 Hz), 3.07 (1H, dd, *J* = 11.5, 6.8 Hz). ^13^C NMR (150 MHz, CDCl_3_) δ (ppm): 159.4, 156.7, 138.0, 130.3, 129.5, 129.2(2C), 128.8, 128.7(2C), 127.6, 127.5, 127.0, 125.4, 124.6(2C), 124.2, 123.8, 116.9(2C), 115.0, 87.5, 84.1, 59.4, 55.4, 52.9, 44.8, 33.3. IR spectrum (KBr), υ/cm^−1^: 2222.9 (-C≡C-). HRMS (MALDI+) *m*/*z* calcd for C_28_H_23_ClN_2_O in form of [M + H]^+^ ion 439.1577, found: 439.1558.

**10-chloro-*2*-methyl-1-phenylethynyl-1,2,3,4-tetrahydrobenzo[*b*][1,6]naphthyridine (5c).** Oil, yield 32%. ^1^H NMR (600 MHz, CDCl_3_) δ(ppm): 8.23 (1H, d, *J* = 8.4 Hz), 8.01 (1H, d, *J* = 8.3 Hz), 7.72 (1H, t, *J* = 7.6 Hz), 7.58 (1H, t, *J* = 7.6 Hz), 7.39 (2H, dd, *J* = 7.4, 2.2 Hz), 7.25 (3H, m), 5.30 (1H, s), 3.41 (1H, ddd, *J* = 17.4, 11.7, 7.3 Hz), 3.28 (1H, td, *J* = 11.8, 4.2 Hz), 3.18 (1H, dd, *J* = 17.3, 4.1 Hz), 2.98 (1H, dd, *J* = 12.0, 7.3 Hz), 2.69 (3H, s). IR spectrum (KBr), υ/cm^−1^: 2225.8 (-C≡C-). HRMS (MALDI+) *m*/*z* calcd for: C_21_H_17_ClN_2_ in form of [M + H]^+^ ion 333.1159, found: 333.1142.

**10-chloro-2-methyl-1-[3-(methoxyphenyl)ethynyl]-1,2,3,4-tetrahydrobenzo[*b*][1,6]naphthyridine (5d).** Oil, yield 85%. ^1^H NMR (600 MHz, CDCl_3_) δ (ppm): 8.23 (1H, d, *J* = 8.5 Hz), 8.04 (1H, d, *J* = 8.4 Hz), 7.73 (1H, t, *J* = 7.7 Hz), 7.59 (1H, t, *J* = 7.7 Hz), 7.17 (1H, t, *J* = 8.0 Hz), 6.99 (1H, d, *J* = 7.7 Hz), 6.91 (1H, s), 6.84 (1H, dd, *J* = 8.5, 2.6 Hz), 5.35 (1H, s), 3.75 (3H, s), 3.49 (1H, ddd, *J* = 18.2, 11.8, 7.3 Hz), 3.34 (1H, td, *J* = 11.9, 4.5 Hz), 3.23 (1H, dd, *J* = 17.5, 4.3 Hz), 3.05 (1H, dd, *J* = 12.1, 7.2 Hz), 2.74 (3H, s). ^13^C NMR (150 MHz, CDCl_3_) δ (ppm): 159.3, 155.4, 147.4, 140.9, 130.6, 129.5, 128.7, 127.2, 125.4, 124.5(2C), 124.2, 123.4, 116.8, 115.2, 88.1, 82.8, 55.4, 54.6, 46.9, 43.1, 32.6. IR spectrum (KBr), υ/cm^−1^: 2216.1 (-C≡C-). HRMS (MALDI+) *m*/*z* calcd for: C_22_H_19_ClN_2_O in form of [M + H]^+^ ion 363.1264, found: 363.1281.

**10-chloro-2-methyl-1-[4-(methoxyphenyl)ethynyl]-1,2,3,4-tetrahydrobenzo[*b*][1,6]naphthyridine (5e).** Oil, yield 88%. ^1^H NMR (600 MHz, CDCl_3_) δ(ppm): 8.23 (1H, d, *J* = 8.3 Hz), 8.01 (1H, d, *J* = 8.4 Hz), 7.72 (1H, ddd, *J* = 8.3, 6.9, 1.4 Hz), 7.58 (1H, ddd, *J* = 8.2, 6.8, 1.2 Hz), 7.33 (2H, d, *J* = 8.9 Hz), 6.79 (2H, d, *J* = 8.9 Hz), 5.28 (1H, s), 3.77 (3H, s), 3.46–3.34 (1H, m), 3.28 (1H, td, *J* = 11.7, 4.3 Hz), 3.18 (1H, dd, *J* = 17.4, 3.4 Hz), 2.97 (1H, dd, *J* = 11.9, 7.3 Hz), 2.68 (3H, s). ^13^C NMR (150 MHz, CDCl_3_) δ (ppm): 159.7, 156.1, 147.6, 140.4, 133.3(2C), 130.2, 128.9, 127.6, 126.9, 125.4, 124.3, 114.9, 113.9(2C), 87.5, 82.3, 55.4, 54.7, 46.9, 43.2, 33.1. IR spectrum (KBr), υ/cm^−1^: 2222.3 (-C≡C-). HRMS (MALDI+) *m*/*z* calcd for: C_22_H_19_ClN_2_O in form of [M + H]^+^ ion 363.1264, found: 363.1279.

**2-benzyl-10-chloro-1-{[4-(trifluoromethyl)phenyl]ethynyl}-1,2,3,4-tetrahydrobenzo[*b*][1,6]naphthyridine (5f).** Oil, yield 61%. ^1^H NMR (600 MHz, CDCl_3_) δ(ppm): 8.24 (1H, d, *J* = 8.6 Hz), 8.05 (1H, d, *J* = 8.6 Hz), 7.75 (1H, t, *J* = 7.6 Hz), 7.60 (1H, t, *J* = 7.7 Hz), 7.42 (2H, *J* = 8.5 Hz), 7.12 (2H, d, *J* = 8.4 Hz), 5.34 (1H, s), 3.52–3.46 (1H, m), 3.33–3.26 (1H, m), 3.23 (1H, dd, *J* = 17.4, 4.1 Hz), 3.05 (1H, dd, *J* = 12.2, 7.4 Hz), 2.73 (3H, s). ^13^C NMR (150 MHz, CDCl_3_) δ (ppm): 155.5, 149.2, 133.5(2C), 130.6, 128.7, 127.32, 125.4, 124.3(2C), 121.4, 121.3, 120.9(2C), 119.6, 86.8, 84.2, 60.5, 54.6, 47.0, 43.1, 32.6. IR spectrum (KBr), υ/cm^−1^: 2221.2 (-C≡C-). HRMS (MALDI+) *m*/*z* calcd for: C_22_H_16_ClN_2_F_3_ in form of [M + H]^+^ ion 401.1032, found: 401.1041.

**10-chloro-1-[(4-fluorophenyl)ethynyl]-2-methyl-1,2,3,4-tetrahydrobenzo[*b*][1,6]naphthyridine (5g).** Oil, yield 44%. ^1^H NMR (600 MHz, CDCl_3_) δ (ppm): 8.23 (1H, d, *J* = 8.4 Hz), 8.03 (1H, d, *J* = 8.5 Hz), 7.73 (1H, t, *J* = 7.3 Hz), 7.59 (1H, t, *J* = 7.6 Hz), 7.40–7.35 (2H, m), 6.99–6.93 (2H, m), 5.31 (1H, s), 3.45 (1H, ddd, *J* = 17.4, 11.7, 7.3 Hz), 3.29 (1H, td, *J* = 11.8, 4.4 Hz), 3.20 (1H, dd, *J* = 17.4, 4.3 Hz), 3.02 (1H, dd, *J* = 11.9, 7.2 Hz), 2.71 (3H, s). ^13^C NMR (150 MHz, CDCl_3_) δ (ppm): 163.3, 161.6, 155.6, 147.3, 140.5, 133.7, 133.6, 130.2, 128.6, 126.9, 125.1, 124.0, 118.5, 115.5, 115.4, 86.6, 83.0, 54.5, 46.8, 42.9, 32.6. IR spectrum (KBr), υ/cm^−1^: 2227.1 (-C≡C-). IR spectrum (KBr), υ/cm^−1^: 2227.1 (-C≡C-). HRMS (MALDI+) *m*/*z* calcd for C_21_H_16_ClFN_2_ in form of [M + H]^+^ ion 351.1064, found: 351.1049.

**10-chloro-1-[(4-chlorophenyl)ethynyl]-2-methyl-1,2,3,4-tetrahydrobenzo[*b*][1,6]naphthyridine (5h).** Oil, yield 59%. ^1^H NMR (600 MHz, CDCl_3_) δ(ppm): 8.24 (1H, dd, *J* = 8.5, 0.9 Hz), 8.11 (1H, d, *J* = 8.4 Hz), 7.78 (1H, ddd, *J* = 8.3, 6.9, 1.4 Hz), 7.63 (1H, ddd, *J* = 8.3, 6.9, 1.2 Hz), 7.32 (2H, d, *J* = 8.7 Hz), 7.25 (2H, d, *J* = 6.7 Hz), 5.44 (1H, s), 3.67–3.58 (1H, m), 3.45–3.36 (1H, m), 3.32 (1H, dd, *J* = 17.8, 4.4 Hz), 3.17 (1H, m), 2.81 (3H, s). IR spectrum (KBr), υ/cm^−1^: 2223.0 (-C≡C-). HRMS (MALDI+) *m*/*z* calcd for: C_21_H_16_Cl_2_N_2_ in form of [M + H]^+^ ion 367.0769, found: 367.0780.

#### 3.1.4. Synthesis of Compounds **6a**–**f**

The appropriate tetrahydrobenzonaphthyridines **5c**,**d**,**f**,**g** were dissolved in isopropanol at room temperature and cooled for 10 min in the freezer, then a 1.2 equivalent of activated alkyne was added and the mixture was stored in the refrigerator for 1 week. The reaction was controlled by TLC in an ethyl acetate/n-hexane 1:1 system on Silufol plates. The product was separated by column chromatography.

**Methyl (2E)-3-{10-chloro-1-[(4-fluorophenyl)ethynyl]-2-methyl-1,2,3,4-tetrahydrobenzo[*b*][1,6]naphthyridin-1-yl}prop-2-enoate (6a).** Oil, yield 27%. ^1^H NMR (600 MHz, CDCl_3_) δ (ppm): 8.26 (1H, d, *J* = 8.6, 1.4, 0.6 Hz), 7.99 (1H, d, *J* = 8.5, 0.6 Hz), 7.74 (1H, t, *J* = 7.6 Hz), 7.56 (1H, t, *J* = 7.6 Hz), 7.44 (2H, dd *J* = 8.8, 5.3 Hz), 7.00 (2H, t, *J* = 8.7 Hz), 6.83 (1H, d, *J* = 15.6 Hz), 6.50 (1H, d, *J* = 15.6 Hz), 3.76 (3H, s), 3.53–3.49 (1H, m), 3.17–3.12 (2H, m), 3.07–3.03 (1H, m), 2.53 (3H, s). ^13^C NMR (150 MHz, CDCl_3_) δ (ppm): 166.9, 163.8, 162.1, 156.5, 147.5, 142.8, 134.1, 134.1, 131.0, 128.9, 128.7, 127.4, 126.4, 125.2, 124.7, 124.6, 118.8, 116.0, 88.3, 83.46, 63.6, 52.1, 48.0, 40.0, 35.1. IR spectrum (KBr), υ/cm^−1^: 2224.3 (-C≡C-); 1724 (C = O). HRMS (MALDI+) *m*/*z* calcd for: C_25_H_20_ClFN_2_O_2_ in form of [M + H]^+^ ion 435.1276, found: 435.1261.

**Methyl (2E)-3-{10-chloro-1-[(3-methoxyphenyl)ethynyl]-2-methyl-1,2,3,4-tetrahydrobenzo[*b*][1,6]naphthyridin-1-yl}prop-2-enoate (6b).** Oil, yield 22%. ^1^H NMR (600 MHz, CDCl_3_) δ (ppm): 8.27 (1H, d, *J* = 8.1 Hz), 7.99 (1H, d, *J* = 8.3 Hz), 7.73 (1H, t, *J* = 7.6 Hz), 7.57 (1H, t, *J* = 7.7 Hz), 7.21 (1H, t, *J* = 8.0 Hz), 7.07 (1H, d, *J* = 7.6 Hz), 6.98 (1H, s), 6.88 (1H, dd, *J* = 8.2, 2.7 Hz), 6.83 (1H, d, *J* = 15.6 Hz), 6.51 (1H, d, *J* = 15.6 Hz), 3.79 (3H, s), 3.75 (3H, s), 3.51 (1H, ddd, *J* = 17.9, 12.0, 6.1 Hz), 3.20–3.12 (2H, m), 3.05 (1H, ddd, *J* = 11.7, 6.1, 1.9 Hz), 2.53 (3H, s). ^13^C NMR (150 MHz, CDCl_3_) δ (ppm): 166.7, 159.4, 156.3, 147.3, 142.6, 130.7, 130.2, 129.5, 129.4, 128.6, 128.5, 127.1, 128.8, 126.2, 124.9, 124.5, 123.4, 116.9, 115.1, 89.0, 83.3, 63.4, 60.5, 55.4, 51.9, 47.8. IR spectrum (KBr), υ/cm^−1^: 2224.3 (-C≡C-); 1724 (C = O). HRMS (MALDI+) *m*/*z* calcd for C_26_H_23_ClN_2_O_3_ in form of [M + H]^+^ ion 447.1476, found: 447.1460.

**(3E)-4-[10-chloro-2-methyl-1-(phenylethynyl)-1,2,3,4-tetrahydrobenzo[*b*][1,6]naphthyridin-1-yl]but-3-en-2-one (6c).** White crystals, yield 37%, m.p. = 154–155 °C. ^1^H NMR (600 MHz, CDCl_3_) δ (ppm): 8.28 (1H, d, *J* = 8.5 Hz), 8.01 (1H, d, *J* = 8.5 Hz), 7.75 (1H, ddd, *J* = 8.3, 6.8, 1.3 Hz), 7.58 (1H, ddd, *J* = 8.3, 6.8, 1.2 Hz), 7.47 (2H, dd, *J* = 7.4, 1.9 Hz), 7.32–7.30 (3H, m), 6.73 (1H, d, *J* = 16.1 Hz), 6.63 (1H, d, *J* = 16.1 Hz), 3.59–3.47 (1H, m), 3.24–3.11 (2H, m), 3.06 (1H, ddd, *J* = 13.1, 6.1, 2.5 Hz), 2.53 (3H, s), 2.31 (3H, s). ^13^C NMR (150 MHz, CDCl_3_) δ (ppm): 198.6, 156.0, 147.1, 146.2, 142.4, 134.2(2C), 131.8(2C), 130.6, 128.6, 128.5, 128.3(2C), 127.0(2C), 126.0, 124.3, 122.2, 89.3, 82.9, 63.5, 47.7, 39.6, 34.7. IR spectrum (KBr), υ/cm^−1^: 2219.3 (-C≡C-); 1675.5 (C = O). HRMS (MALDI+) *m*/*z* calcd for: C_25_H_21_ClN_2_O in form of [M + H]^+^ ion 401.1421, found: 401.1411.

**(3E)-4-{10-chloro-1-[(3-methoxyphenyl)ethynyl]-2-methyl-1,2,3,4-tetrahydrobenzo[*b*][1,6]naphthyridin-1-yl}but-3-en-2-one (6d).** Oil, yield 34%. ^1^H NMR (600 MHz, CDCl_3_) δ(ppm): 8.28 (1H, ddd, *J* = 8.5, 1.3, 0.6 Hz), 8.01 (1H, dd, *J* = 8.5, 0.6 Hz), 7.75 (1H, ddd, *J* = 8.4, 6.8, 1.4 Hz), 7.58 (1H, ddd, *J* = 8.3, 6.8, 1.2 Hz), 7.21 (1H, ddd, *J* = 8.2, 7.6, 0.4 Hz), 7.07 (1H, ddd, *J* = 7.6, 1.4, 1.0 Hz), 6.97 (1H, dd, *J* = 2.7, 1.3 Hz), 6.88 (1H, ddd, *J* = 8.4, 2.6, 1.0 Hz), 6.72 (1H, d, *J* = 16.1 Hz), 6.62 (1H, d, *J* = 16.1 Hz), 3.79 (3H, s), 3.53 (1H, ddd, *J* = 16.0, 11.6, 6.2 Hz), 3.23–3.11 (2H, m), 3.06 (1H, ddd, *J* = 13.4, 6.2, 2.8 Hz), 2.53 (3H, s), 2.31 (3H, s). ^13^C NMR (150 MHz, CDCl_3_) δ (ppm): 166.8, 159.5, 156.3, 147.3, 142.7, 130.8, 129.5, 128.7, 128.6, 127.1, 126.2, 125.0, 124.6, 124.5, 123.5, 122.1, 116.9, 115.2, 111.8, 89.0, 83.3, 63.4, 60.5, 55.4, 51.9, 47.8. IR spectrum (KBr), υ/cm^−1^: 2219.3 (-C≡C-); 1723.8 (C = O). HRMS (MALDI+) *m*/*z* calcd for: C_26_H_23_ClN_2_O_2_ in form of [M + H]^+^ ion 431.1526, found: 431.1539.

**(3E)-4-(10-chloro-2-methyl-1-{[4-(trifluoromethyl)phenyl]ethynyl}-1,2,3,4-tetrahydrobenzo[*b*][1,6]naphthyridin-1-yl)but-3-en-2-one (6e).** Oil, yield 42%. ^1^H NMR (600 MHz, CDCl_3_) δ (ppm): 8.28 (1H, d, *J* = 8.5 Hz), 8.01 (1H, d, *J* = 8.5 Hz), 7.76 (1H, t, *J* = 7.7 Hz), 7.64–7.55 (1H, m), 7.49 (2H, d, *J* = 8.4 Hz), 7.16 (2H, d, *J* = 8.8 Hz), 6.69 (1H, d, *J* = 16.1 Hz), 6.62 (1H, d, *J* = 16.1 Hz), 3.56–3.50 (1H, m), 3.17 (2H, m), 3.07 (1H, m), 2.52 (3H, s), 2.31 (3H, s). IR spectrum (KBr), υ/cm^−1^: 2220.9 (-C≡C-); 1721.4 (C = O). HRMS (MALDI+) *m*/*z* calcd for: C_26_H_20_ClF_3_N_2_O in form of [M + H]^+^ ion 469.1295, found: 469.1301.

**Methyl 5-(4-chloro-2-ethenylquinolin-3-yl)-4-(4-fluorobenzyl)-1-methyl-1H-pyrrole-3-carboxylate (7).** Oil, yield 3%. ^1^H NMR (600 MHz, CDCl_3_) δ(ppm): 8.07 (1H, d, ddd, *J* = 8.5, 1.4, 0.7 Hz), 7.99 (1H, d, *J* = 8.5 Hz), 7.69 (1H, ddd, *J* = 8.4, 6.9, 1.4 Hz), 7.56 (1H, ddd, *J* = 8.3, 6.9, 1.1 Hz), 7.25 (1H, s), 7.07–7.04 (2H, m), 6.74 (2H, t, *J* = 8.8 Hz), 6.68 (1H, dd, *J* = 16.7, 10.6 Hz), 6.32 (1H, dd, *J* = 16.8, 2.0 Hz), 5.42 (1H, dd, *J* = 10.6, 1.9 Hz), 4.40 (2H, s), 3.60 (3H, s), 3.46 (3H, s). ^13^C NMR (150 MHz, CDCl_3_) δ (ppm): 165.0, 155.6, 133.7, 132.0, 132.0, 130.9, 130.9, 130.3(2C), 129.9, 128.2(2C), 127.7(2C), 126.9, 125.7, 124.5, 124.5, 124.5, 122.4, 114.5, 114.3, 51.0, 35.5, 26.8. HRMS (MALDI+) *m*/*z* calcd for: C_25_H_20_ClFN_2_O_2_ in form of [M + H]^+^ ion 435.1276, found: 435.1268.

#### 3.1.5. Synthesis of 2-Benzyl-10-chloro-1-(indol-3-yl)-1,2,3,4-tetrahydrobenzo[*b*][1,6]naphthyridines **8a**–**d**

A solution of **3a** (0.5 g) in 10 mL of THF was cooled to 0 °C, then a 1.2 equivalent excess of DIAD (diisopropylazodicarboxylate) was added. The mixture was stirred at room temperature for 1 h. After cooling it again to 0 °C, a 1.5 equivalent excess of the appropriate indole was added. The reaction was stirred at room temperature and controlled by TLC in an ethyl acetate/hexane (1:1) system on Silufol plates. The product was separated by column chromatography.

**2-benzyl-10-chloro-1-(5-methoxy-1H-indol-3-yl)-1,2,3,4-tetrahydrobenzo[*b*][1,6]naphthyridine (8a).** Yellow foamed oil, yield 61%. ^1^H NMR (600 MHz, CDCl_3_) δ (ppm): 8.17 (1H, d, *J* = 7.6 Hz), 8.08 (1H, d, *J* = 8.4 Hz), 8.03 (1H, s), 7.74 (1H, ddd, *J* = 8.4, 6.9, 1.3 Hz), 7.57 (1H, ddd, *J* = 8.1, 6.9, 1.0 Hz), 7.44 (2H, d, *J* = 7.2 Hz), 7.37 (2H, t, *J* = 7.5 Hz), 7.33–7.29 (1H, m), 7.20 (1H, d, *J* = 8.7 Hz), 7.01 (1H, d, *J* = 2.4 Hz), 6.84 (1H, dd, *J* = 8.7, 2.5 Hz), 6.36 (1H, d, *J* = 2.5 Hz), 5.76 (1H, s), 4.00 (1H, d, *J* = 13.2 Hz), 3.79 (3H, s), 3.65 (1H, d, *J* = 13.2 Hz), 3.44–3.33 (2H, m), 3.09–3.05 (1H, m), 3.03–2.99 (1H, m). ^13^C NMR (150 MHz, CDCl_3_) δ (ppm): 157.5, 154.2, 147.5, 141.9, 139.2, 131.4, 130.0(2C), 129.6, 129.3(2C), 128.8, 128.6(2C), 127.8, 127.4, 126.9(2C), 125.0, 124.2, 115.3, 112.9, 111.9, 101.3, 57.5, 55.9, 42.5, 29.1. HRMS (MALDI+) *m*/*z* calcd for: C_28_H_24_ClN_3_O in form of [M + H]^+^ ion 454.1686, found: 454.1669.

**2-benzyl-10-chloro-1-(1H-indol-3-yl)-1,2,3,4-tetrahydrobenzo[*b*][1,6]naphthyridine (8b).** Compound was isolated in mixture with hydrazine **9**. Yellow foamed oil, yield 48%. ^1^H NMR 1H NMR (600 MHz, CDCl_3_) δ (ppm): 8.31 (1H, s), 8.22 (1H, s), 8.16 (1H, d, *J* = 8.3 Hz), 8.08 (1H, d, *J* = 8.4 Hz), 7.73 (1H, t, *J* = 7.7 Hz), 7.62 (1H, d, *J* = 8.0 Hz), 7.56 (1H, t, *J* = 7.6 Hz), 7.41 (1H, d, *J* = 7.5 Hz), 7.38–7.33 (2H, m), 7.34–7.28 (3H, m), 7.19 (1H, t, *J* = 7.6 Hz), 7.15–7.09 (2H, m), 6.42 (1H, s), 5.84 (1H, s), 4.01 (1H, d, *J* = 13.3 Hz), 3.61 (1H, d, *J* = 13.2 Hz), 3.39 (1H, ddd, *J* = 18.4, 11.8, 7.3 Hz), 3.26 (1H, td, *J* = 12.5, 5.0 Hz), 3.10 (1H, dd, *J* = 17.9, 4.8 Hz), 2.94 (1H, dd, *J* = 13.1, 7.3 Hz). HRMS (MALDI+) *m*/*z* calcd for: C_27_H_22_ClN_3_ in form of [M + H]^+^ ion 424.1581, found: 424.1591.

**2-benzyl-10-chloro-1-(5-chloro-1H-indol-3-yl)-1,2,3,4-tetrahydrobenzo[*b*][1,6]naphthyridine (8c).** Compound was isolated in mixture with hydrazine **9**. Yellow foamed oil, yield 51%**.**
^1^H NMR (600 MHz, CDCl_3_) δ (ppm): 8.34 (1H, s), 8.17–8.16 (2H, m), 8.08 (1H, d, *J* = 8.4 Hz), 7.75 (1H, ddd, *J* = 8.3, 6.7, 1.4 Hz), 7.61–7.56 (2H, m), 7.54 (1H, d, *J* = 2.1 Hz), 7.42–7.39 (5H, m), 7.35–7.33 (2H, m), 7.24 (2H, d, *J* = 8.6 Hz), 7.14 (2H, dd, *J* = 8.6, 2.0 Hz), 6.38 (1H, d, *J* = 2.4 Hz), 5.73 (1H, s), 3.97 (1H, d, *J* = 13.1 Hz), 3.62 (1H, d, *J* = 13.1 Hz), 3.39 (1H, ddd, *J* = 18.4, 12.0, 7.2 Hz), 3.29 (1H, td, *J* = 12.7, 4.9 Hz), 3.06 (1H, dd, *J* = 17.9, 4.8 Hz), 3.01 (1H, dd, *J* = 13.4, 7.1 Hz). HRMS (MALDI+) *m*/*z* calcd for: C_27_H_21_Cl_2_N_3_ in form of [M + H]^+^ ion 458.1191, found: 458.1183.

**2-benzyl-10-chloro-1-(5-bromo-1H-indol-3-yl)-1,2,3,4-tetrahydrobenzo[*b*][1,6]naphthyridine (8d).** Compound was isolated in mixture with hydrazine **9**. Yellow foamed oil, yield 69%. ^1^H NMR (600 MHz, CDCl_3_) δ (ppm): 8.18 (1H, s), 8.17 (1H, dd, *J* = 8.3, 1.3 Hz), 8.08 (1H, d, *J* = 8.5 Hz), 7.77–7.72 (2H, m), 7.69 (1H, s), 7.58 (1H, ddd, *J* = 8.3, 6.8, 1.2 Hz), 7.42 (3H, d, *J* = 4.4 Hz), 7.37–7.33 (1H, m), 7.27 (1H, d, *J* = 1.9 Hz), 7.19 (1H, d, *J* = 8.6 Hz), 6.34 (1H, s), 5.72 (1H, s), 3.96 (1H, d, *J* = 13.0 Hz), 3.62 (1H, d, *J* = 13.0 Hz), 3.39 (1H, ddd, *J* = 18.1, 12.0, 7.2 Hz), 3.30 (1H, td, *J* = 12.7, 4.9 Hz), 3.06 (1H, dd, *J* = 17.9, 4.8 Hz), 3.02 (1H, dd, *J* = 13.3, 7.1 Hz). HRMS (MALDI+) *m*/*z* calcd for: C_27_H_21_BrClN_3_ in form of [M + H]^+^ ion 502.0686, found: 502.0678.

#### 3.1.6. Synthesis of 1-[2-Benzyl-6-chloro-5-(indol-3-yl)-2,5-dihydro-1H-azepino[3,4-b]quinolin-4-yl]ethenones **10a**,**b**

The appropriate compounds **8a**,**d** were dissolved in isopropanol at room temperature and cooled for 10 min in the freezer, then a 1.2 equivalent of activated alkyne was added and the mixture was stored in the refrigerator for 1 week. The reaction was controlled by TLC in an ethyl acetate/hexane 1:1 system on Silufol plates. The compound spontaneously fell out of the reaction mass in the form of crystals and was released by filtration.

**1-[(4*E*)-3-benzyl-7-chloro-6-(5-methoxy-1*H*-indol-3-yl)-1,2,3,6-tetrahydroazocino[4,5-*b*]quinolin-5-yl]ethanone (10a).** White crystals, yield 87%, m.p. = 147–149 °C. ^1^H NMR (600 MHz, CDCl_3_) δ (ppm): 8.33 (1H, s), 8.30 (1H, dd, *J* = 8.3, 1.3 Hz), 7.94 (1H, d, *J* = 8.4 Hz), 7.76 (1H, s), 7.72 (1H, ddd, *J* = 8.4, 6.9, 1.4 Hz), 7.61 (1H, ddd, *J* = 8.3, 6.9, 1.2 Hz), 7.21 (1H, s), 7.18 (1H, d, *J* = 8.8 Hz), 7.15–7.10 (1H, m), 7.03 (2H, t, *J* = 7.7 Hz), 6.74 (1H, dd, *J* = 8.8, 2.5 Hz), 6.71 (2H, d, *J* = 7.4 Hz), 6.61 (1H, s), 4.31 (1H, d, *J* = 15.0 Hz), 4.22 (1H, d, *J* = 15.0 Hz), 4.22–4.18 (1H, m), 3.47 (3H, s), 3.30 (1H, ddd, *J* = 16.9, 10.9, 7.4 Hz), 3.17–3.11 (2H, m), 2.42 (3H, s). ^13^C NMR (150 MHz, CDCl_3_) δ (ppm): 195.0, 174.5, 171.2, 160.4, 156.5, 153.6, 136.0, 132.3, 132.0, 129.7, 128.7(2C), 128.6, 127.9(2C), 127.3(2C), 127.0, 126.3, 125.9, 125.1(2C), 122.4, 119.3, 112.0, 101.3, 61.8, 60.4, 55.4, 51.0, 39.8, 36.5. IR spectrum (KBr), υ/cm^−1^: 1695.7 (C = O). HRMS (MALDI+) *m*/*z* calcd for: C_32_H_28_ClN_3_O_2_ in form of [M + H]^+^ ion 522.1948, found: 522.1956.

**1-[(4*E*)-3-benzyl-7-chloro-6-(5-bromo-1*H*-indol-3-yl)-1,2,3,6-tetrahydroazocino[4,5-*b*]quinolin-5-yl]ethanone (10b).** White crystals, yield 82%, m.p. =136–137 °C. ^1^H NMR (600 MHz, CDCl_3_) δ (ppm): 8.32 (1H, s), 8.30 (1H, d, *J* = 8.4 Hz), 7.94 (1H, d, *J* = 8.8 Hz), 7.78–7.69 (2H, m), 7.62 (1H, t, *J* = 7.6 Hz), 7.24 (1H, s), 7.22–7.14 (2H, m), 7.15 (1H, t, *J* = 7.4 Hz), 7.06 (2H, t, *J* = 7.6 Hz), 6.72 (2H, d, *J* = 7.4 Hz), 6.61 (1H, s), 4.32 (1H, d, *J* = 15.0 Hz), 4.24 (1H, d, *J* = 15.0 Hz), 4.16–4.09 (1H, m), 3.31 (1H, ddd, *J* = 16.7, 10.6, 7.1 Hz), 3.17–3.11 (2H, m), 2.42 (3H, s). ^13^C NMR (150 MHz, CDCl_3_) δ (ppm): 194.7, 159.9, 156.1, 146.4, 143.3, 135.8, 135.3, 131.7, 129.7, 128.7, 128.6(2C), 127.8(2C), 127.6, 127.2(2C), 127.0, 125.8, 125.1, 124.9, 122.6, 121.9, 119.7, 112.7, 112.6, 61.8, 50.7, 40.0, 35.9, 25.4. IR spectrum (KBr), υ/cm^−1^: 1700.3 (C = O). HRMS (MALDI+) *m*/*z* calcd for: C_31_H_25_BrClN_3_O in form of [M + H]^+^ ion 570.0948, found: 570.0969.

## 3.2. Biochemical Assays

### MAO Inhibition

All reagents were purchased from Sigma Aldrich (Milan, Italy). The fluorometric assay was performed as previously described [17] using human recombinant enzymes from baculovirus-infected insect cells, following the formation of fluorescing 4-hydroxyquinoline from the MAO substrate, kynuramine. Assays were performed in triplicate in 96-well plates (Greiner Bio-One GmbH, Frickenhausen, Germany) using an Infinite M1000 multiplate reader (Tecan, Cernusco sul Naviglio (MI), Italy). Results were expressed as mean ± SEM. IC_50_ values were obtained by nonlinear regression using Prism software (GraphPad Prism version 5.00 for Windows, GraphPad Software, San Diego, CA, USA).

## 4. Conclusions

As a major outcome of this study, novel functionalized 2-alkyl-10-chloro-1,2,3,4-tetrahydrobenzo[*b*][1,6]naphthyridines **3** were synthesized for the first time, specifically 1-phenylethynyl derivatives **5** and 1-indol-3-yl derivatives **8**. Moreover, the interaction of these compounds with activated alkynes was studied, revealing that the substituent in the first position played a key role in these reactions and either Stevens rearrangement products or azocino[4,5-*b*]quinolines were formed.

The 1-phenylethynyl derivatives **5c**–**h** were discovered as MAO inhibitors, showing selectivity toward the human MAO B isoform and potency in the low micromolar range. In particular, the 4-F derivative **5g** achieved an IC_50_ of 1.35 μM in vitro, which was almost equipotent with pargyline (IC_50_ 2.69), a known MAO B irreversible inhibitor that was taken as the positive control. MAO B inhibitors are typically used in the treatment of early symptoms of PD [47], while their efficacy in decreasing oxidative stress may provide neuroprotective effects in the treatment of AD [48]. In this context, compound **5g** deserves further optimization studies for improving its pharmacological potential as an effective agent for the treatment of neurodegenerative syndromes.

## Data Availability

All data presented in this study are available in the article and in Appendix A.

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
