# Peer review of "Synthesis of Novel Benzo[*b*][1,6]naphthyridine Derivatives and Investigation of Their Potential as Scaffolds of MAO Inhibitors"

_molecules, 2023, doi:10.3390/molecules28041662_

Round 1

Reviewer 1 Report

The paper describes the synthesis of novel benzo[b][1,6]naphthyridine derivatives and the investigation of their potential as scaffolds of MAO inhibitors. The work is interesting and the presentation is good. Most of the compounds prepared are novel and they are all fully characterised.

Based on that I suggest that the paper is accepted after the following are addressed.

Corrections/comments:

- Scheme 5. The mechanism for the step going from compound B to compound 7 is incomplete. Please add all the curly arrows.

- Line 155: I am not aware of a Hoffmann splitting reaction. Maybe the authors mean a Hoffmann elimination?

- Experimental: While IR data are reported for some compounds, they are missing for others. Moreover, only one or two absorptions are recorded in each case. Please report the full IR data for all novel compounds.

- Compound 3b is reported in the patent literature and is prepared in a very similar proceedure. Please compare the characterisation data and add the reference.

- The supporting information file needs to be improved. Please have one spectrum per page in landscape configuration in order for better clarity. Moreover, please provide zoomed images for all regions where the peaks are not clearly visible.

Author Response

Reviewer #1:

(the reviewer comments are given in italic)

  • Scheme 5. The mechanism for the step going from compound B to compound 7 is incomplete. Please add all the curly arrows.

The scheme was corrected

  • Line 155: I am not aware of a Hoffmann splitting reaction. Maybe the authors mean a Hoffmann elimination?

You are absolutely right, we have corrected the text accordingly

  • Experimental: While IR data are reported for some compounds, they are missing for others. Moreover, only one or two absorptions are recorded in each case. Please report the full IR data for all novel compounds.

We have added IR data for all novel compounds

  • Compound 3b is reported in the patent literature and is prepared in a very similar proceedure. Please compare the characterisation data and add the reference.

Thank you for your careful attention to our manuscript, this patent will be very useful for us. WE are sorry that have not found it by ourselves.  We have compared the data and it corresponds to our one. The reference has been added.

  • The supporting information file needs to be improved. Please have one spectrum per page in landscape configuration in order for better clarity. Moreover, please provide zoomed images for all regions where the peaks are not clearly visible.

The necessary corrections have been done.

Reviewer 2 Report

In the manuscript, 2-alkyl-10-chloro-1,2,3,4-tetrahydrobenzo[b][1,6]naphthyridines were prepared and their reactivities were further studied. Novel derivatives of the tricyclic scaffolds, including 1-phenylethynyl, 1-indol-3-yl, and azocino[4,5-b]quinoline derivatives, were obtained. Bioassay was carried out and the results indicated that the newly synthesized derivatives 5c-h proved to be MAO B inhibitors with potency in the low micromolar range. It can be accepted for publication after a minor revision.

1.     For the reaction in Scheme 5, the yields are generally low (22-42%), showing poor potential applicability. On the basis of authors’ comments, competitive Stevens rearrangement and Hoffmann elimination occurred. It is well-known that both reactions required basic conditions, please try to add some appropriate base to improve the yields, making the reaction more useful.

2.     It would be better for readers to draw a detailed formation mechanisms for the formation of 6 from A and for the formation of 7 from 6 stepwise.

3.     On the other hand, it would be better to add some discussion why the Stevens rearrangement takes place on the carbon attached with acetylenyl substituent rather than the other side.

4.     English should be improved. Such as

Provide yields for the reactions in Scheme 2.

anthranilic acid 1a-c  -  anthranilic acids 1a-c

derivative 4a,b was obtained  -  derivatives 4a,b were obtained

reaction of compound 3a,b,d with activated alkynes  -  reactions of compounds 3a,b,d with activated alkynes

reaction of compound 5c,d,f,g with activated alkynes -  reactions of compounds 5c,d,f,g with activated alkynes

compound 8b-d was  -  compounds 8b-d were

Author Response

(the reviewer comments are given in italic)

  • For the reaction in Scheme 5, the yields are generally low (22-42%), showing poor potential applicability. On the basis of authors’ comments, competitive Stevens rearrangement and Hoffmann elimination occurred. It is well-known that both reactions required basic conditions, please try to add some appropriate base to improve the yields, making the reaction more useful;

Thank you very much for your recommendations, we will definitely use them in future. In our work we based on the assumption that the starting benzonaphthyridines contain sufficiently basic pyridine-type nitrogen atom which can act as a base.

  • It would be better for readers to draw a detailed formation mechanisms for the formation of 6 from A and for the formation of 7 from 6 stepwise;

The scheme has been redrawn.

  • On the other hand, it would be better to add some discussion why the Stevens rearrangement takes place on the carbon attached with acetylenyl substituent rather than the other side.

The formation of ylide B occurs as a result of the elimination of proton from the C-1 position and does not involve C-3 because of presence of electron-accepting aromatic and triple-bond fragments. So, we suppose that C-1 hydrogen is more acidic than C-3.

  • English should be improved. Such as

Provide yields for the reactions in Scheme 2.

anthranilic acid 1a-c  -  anthranilic acids 1a-c

derivative 4a,b was obtained  -  derivatives 4a,b were obtained

reaction of compound 3a,b,d with activated alkynes  -  reactions of compounds 3a,b,d with activated alkynes

reaction of compound 5c,d,f,g with activated alkynes -  reactions of compounds 5c,d,f,g with activated alkynes

compound 8b-d was  -  compounds 8b-d were

The mistakes were corrected, thank you.

Hope now you can consider our manuscript for publication.

Sincerely,

Prof. Dr. Leonid G. Voskressensky